# A Quantum Mechanical MP2 Study of the Electronic Effect of Nonplanarity on the Carbon Pyramidalization of Fullerene C_60_

**DOI:** 10.3390/nano14191576

**Published:** 2024-09-29

**Authors:** Yuemin Liu, Yunxiang Gao, Tariq Altalhi, Di-Jia Liu, Boris I. Yakobson

**Affiliations:** 1Department of Chemistry and Physics, Prairie View A&M University, Prairie View, TX 77446, USA; 2Department of Chemistry, Rice University, Houston, TX 77005, USA; 3Chemistry Department, Taif University, Taif 21974, Saudi Arabia; ta.altalhi@tu.edu.sa (T.A.); biy@rice.edu (B.I.Y.); 4Chemical Science & Engineering Division, Argonne National Laboratory, Lemont, IL 60439, USA; djliu@anl.gov; 5Department of Materials Science and NanoEngineering, Rice University, Houston, TX 77005, USA

**Keywords:** fullerene C_60_, MP2 method, delocalization, bond–antibond interaction, correlation, HOMO-LUMO gap

## Abstract

Among C_60_’s diverse functionalities, its potential application in CO_2_ sequestration has gained increasing interest. However, the processes involved are sensitive to the molecule’s electronic structure, aspects of which remain debated and require greater precision. To address this, we performed structural optimization of fullerene C_60_ using the QM MP2/6–31G* method. The nonplanarity of the optimized icosahedron is characterized by two types of dihedral angles: 138° and 143°. The 120 dihedrals of 138° occur between two hexagons intersecting at C–C bonds of 1.42 Å, while the 60 dihedrals of 143° are observed between hexagons and pentagons at C–C bonds of 1.47 Å. NBO analysis reveals less pyramidal sp^1.78^ hybridization for carbons at the 1.42 Å bonds and more pyramidal sp^2.13^ hybridization for the 1.47 Å bonds. Electrostatic potential charges range from −0.04 a.u. to 0.04 a.u. on the carbon atoms. Second-order perturbation analysis indicates that delocalization interactions in the C–C bonds of 1.42 Å (143.70 kcal/mol) and 1.47 Å (34.98 kcal/mol) are 22% and 38% higher, respectively, than those in benzene. MP2/Def2SVP calculations yield a correlation energy of 13.49 kcal/mol per electron for C_60_, slightly higher than the 11.68 kcal/mol for benzene. However, the results from HOMO-LUMO calculations should be interpreted with caution. This study may assist in the rational design of fullerene C_60_ derivatives for CO_2_ reduction systems.

## 1. Introduction

Fullerene C_60_, also known as Buckminsterfullerene, was first discovered by Kroto et al. in 1985, who earned the Nobel Prize in Chemistry in 1996 [1]. This carbon allotrope was made accessible for practical applications through the synthetic technique developed by Kratschmer et al. [2]. Due to its unique structural, chemical, and mechanical properties, fullerene has been functionalized for numerous applications, including cosmetics, food additives, materials science, fuel cells, electrochemistry, and drug delivery [3,4,5,6]. The structural and electronic properties of fullerene C_60_ have been extensively investigated using experimental [2,7,8,9,10,11,12,13,14,15,16,17,18,19], classical mechanical [16,20,21,22,23,24,25,26,27,28,29,30,31], and quantum mechanical methods [7,16,19,29,32,33,34,35,36,37,38,39,40,41,42,43,44,45,46,47,48,49,50,51,52,53,54,55,56,57,58,59,60,61]. With its delocalized conjugated π system on a nonplanar spheroidal geometry, there has been ongoing debate about its aromaticity [62]. The high positive heat of formation suggests that fullerene C_60_ is anti-aromatic rather than aromatic [12]. Electron correlation plays a crucial role in calculating the low-lying excited T1u states of C_60_’s delocalized conjugated π bonds [43], and the HOMO-LUMO energy gap has been closely linked to the molecule’s reactivity [42]. Most quantum mechanical studies on fullerene C60 have utilized hybrid density functional theory (DFT). Electron correlation effects in the fullerene cage have revealed different local minimum geometries for C_20_, with varying DFT functionals showing differences in predicting the relative ordering of C_20_ isomer energies [63]. Moreover, hybrid DFT performance has been found to degrade with increased pentagon–pentagon strain energy and the percentage of Hartree–Fock (HF) exchange in the DFT method [39,64]. The QM MP2 natural bond orbitals (NBO) analysis offers valuable insights into the structural properties of fullerene in terms of molecular orbitals [65,66,67,68]. Fullerene-based CO_2_ sequestration has shown potential for converting environmental CO_2_ into carbohydrates [17,58,59]. Characterizing fullerene C_60_ using MP2-based NBO analysis can deepen our understanding of the structural and electronic properties of its nonplanar carbon cage, facilitating the development of fullerene C_60_ derivatives, including nanobuds, for applications such as CO_2_ reduction [32].

## 2. Materials and Methods

The structures of fullerene C_60_ and benzene were initially constructed and verified using visual molecular dynamics (VMD) [69] as the starting point for structural optimization. The optimized geometry is shown in Figure 1a, and the representative atoms C1, C2, and C6 are indicated in Figure 1b (the cartesian coordinates of the optimized structure for fullerene C_60_ are available in the Appendix A). Computational simulations were performed using coupled cluster single and double with perturbative triples (CCSD(T)), second-order Møller–Plesset perturbation theory (MP2), primarily MP2/6–31G*, and density functional theory (DFT) methods through the Gaussian 16 software package by Frisch et al. [70]. The DFT optimization was conducted using Becke’s three-parameter hybrid functional for exchange and the Lee–Yang–Parr correlation functional (B3LYP) [71,72], as well as the Becke–Perdew (BP86) [73,74,75] functional. For single-point calculations, the 3–21G*, 6–31G*, and two Karlsruhe basis sets were employed: the valence double-ζ basis set with polarization functions (def2-SVP) and the valence triple-ζ basis set with polarization functions on main-group elements (def2-TZVP) [76]. 

## 3. Results

### 3.1. Bond Length, Bond Angle, and Dihedral Angles of Fullerene

The structural optimization of fullerene C_60_ was performed using the MP2/6–31G* method. The bond lengths, bond angles, dihedral angles, and Merz–Kollman (MK) electrostatic partial charges (ESP) of the optimized geometry are summarized and depicted in Figure 2a, Figure 2b, Figure 3a and Figure 3b, respectively. A comparison of atomic partial charges for the carbon atoms in fullerene C_60_ is provided in Table 1. As shown in Figure 1a, the optimized structure features two types of C=C bonds: 30 double bonds with a length of 1.42 Å, and 60 bonds with a length of 1.47 Å. The C1=C6 double bonds shared between two hexagons have a bond length of 1.42 Å, while the C1=C2 bonds of 1.47 Å connect a hexagon and a pentagon. These bond lengths align well with the experimental value of 1.45 Å reported by David et al. [26]. Both bond lengths are slightly longer than the 1.39 Å typical for benzene but shorter than the 1.54 Å of typical C-C σ single bonds. This suggests that the bonds in fullerene C_60_ are somewhat similar to the conjugated π bonds in benzene but with fewer degrees of conjugation. The curvature of the fullerene surface influences the C-C bonding, as the σ bonds span a non-planar surface crossing both five-membered and six-membered rings. The curvature affects the s and p orbital character, resulting in intermediate hybridization and corresponding bond lengths [9]. In Figure 2b, it is evident that there are 60 bond angles of 108° in the icosahedral symmetry, corresponding to the 12 pentagons, with each pentagon having five bond angles of 108°. Additionally, there are 120 bond angles of 120° corresponding to the 20 hexagonal rings that make up the C_60_ framework.

Dihedral angles reflect the relative orientations or curvature of the five-membered and six-membered rings in fullerene C_60_. Figure 3a shows that fullerene C_60_ exhibits two types of dihedral angles: those shared between two hexagons and those between a hexagon and a pentagon. The dihedral angle between two hexagons is 138°, while the hexagon-pentagon intersection exhibits an angle of 143°. These results are consistent with previous findings by Schein et al. [77]. The negative sign of a dihedral angle indicates the same numerical value but in the opposite direction, defined by different sets of atoms from the two rings. For instance, the dihedral angles C7–C1–C6–C17 at 138° and C2–C1–C6–C5 at −138° refer to identical hexagons in Figure 3a. Additionally, the 30 dihedral angles of 138° correlate with the 30 C=C bonds of 1.42 Å in fullerene C_60_. Similarly, the presence of 60 dihedral angles of 143° is consistent with the 60 C=C bonds of 1.47 Å. The most widely used electrostatic potential (ESP) scheme defines atomic partial charges by fitting a classical Coulomb model to the quantum mechanical electrostatic field [78]. As shown in Table 1, the ESP charges for C1, C6, and C3 are −0.031, 0.009, and −0.035 a.u., respectively. In contrast, the NBO scheme assigns charges of −0.00002, 0.00001, and 0.00001 a.u. to C1, C6, and C2, respectively, while the Mulliken charge assignments give lower values of 0.000011, −0.000022, and −0.000049 a.u. for the same atoms. Lastly, Hirshfeld and CM5 partial charge assignments yield values of −0.00002, 0.000023, and −0.000022 a.u. for C1, C6, and C2, respectively. As seen in Table 1 and Figure 3b, the ESP charges of carbon atoms in fullerene C_60_ range from −0.04 a.u. to 0.04 a.u. Although all five partial charge definitions have a median value of 0.00 a.u., the ESP partial charges are magnitudes higher (by a factor of 1000) than those calculated using NBO, Hirshfeld, Mulliken, and CM5 charge schemes for fullerene C_60_. These significant discrepancies in partial charge assignments will be further investigated in future work.

### 3.2. Natural Bond Analysis

Natural bond analyses were conducted for the optimized structures of fullerene C_60_ and benzene, focusing on the C1–C6 and C1–C2 bonds using the MP2/6–31G* and CCSD(T)/aug-cc-pVTZ methods [68]. The directionality of natural hybrid orbitals (NHOs) is depicted in Figure 1b and compared between fullerene C_60_ and benzene in Table 2. Natural bond coefficients and hybridization are described in Figure 4. The extent of delocalization interactions for the C1–C2, C1–C6 (σ), and C1–C6 (π) bonds is presented in Table 3, while the delocalization energies of the major natural bonds are summarized in Table 4 and illustrated in Figure 4b. The use of localized NBOs offers conceptual simplicity and flexibility over canonical molecular orbitals (CMOs) [68]. NHOs represent a group of natural localized orbitals (NLOs) that transition from atomic basis orbitals to molecular orbitals and are expressed as linear combinations of natural atomic orbitals (NAOs). The NHO orientation is described using azimuthal (Φ) and polar (θ) angles, defining the directionality of the p-component (Figure 1b). As shown in Table 2, planar benzene exhibits a polar angle (θ) deviation of 4.1° for both the C1–C2 and C1–C6 σ bonds (see Figure 1). In contrast, the NHOs in fullerene C_60_ deviate from the center line between the nuclei by 13.1° and 15.1°, respectively. This indicates that the C1–C2 and C1–C6 σ bonds in fullerene C_60_ bend 9° and 11° more from the center line compared to the corresponding bonds in benzene. Furthermore, the NHOs for the C1–C6 π bond in fullerene C_60_ exhibit a polar angle (θ) of 75.8° from the center line, compared to the typical 90° in benzene. This 14.2° deviation reflects the impact of structural non-planarity on the C1–C6 π bond in fullerene C_60_. Tracking the bond bending due to pyramidalization provides valuable insight into the reactivity and stability of crosslinked nanobuds.

As shown in Figure 4, our calculations reveal that the hybrids at C1 and C6 in the C1–C6 NBO of fullerene C_60_ exhibit sp^1.78^ hybridization, slightly smaller than the sp^1.84^ hybridization found in benzene. Meanwhile, the hybrids at C1 and C2 in the C1–C2 NBO display sp^2.13^ hybridization, indicating more pyramidalization compared to the sp^1.84^ hybridization in the C1–C6 bond of benzene. This suggests that the C1–C6 bond in fullerene is similar to the “1.5 bond” found in benzene, whereas the C1–C2 bond resembles a single bond [9]. Bond–antibond interactions were evaluated using second-order perturbative calculations on the NBO basis. The C1–C6 orbital experiences delocalization interactions with various unoccupied orbitals, including non-Lewis core electron pairs, valence lone pairs, Rydberg orbitals, and antibonding orbitals. As indicated in Table 3, both the C1–C6 σ and π orbitals have 4 geminal and 8 vicinal interactions, compared to 2 geminal and 4 vicinal interactions for C1–C6 in benzene. Among these interactions, the most significant involves the less pyramidalized sp^1.78^ bond–antibond interactions in the C1–C6 π bond of fullerene C_60_. Our results, shown in Figure 5 and Table 4, indicate that the strongest delocalization provides an interaction energy of 25.66 kcal/mol between the C1–C6 π bond and four other π antibonds (C2–C3, C4–C5, C7–C18, and C7–C19) in fullerene C_60_. Including additional delocalizations, the total delocalization energy for the C1–C6 π bonds amounts to 115.54 kcal/mol, while the 12 delocalizations involving the C1–C6 σ bond contribute 28.16 kcal/mol. Altogether, the C1–C6 σ and π bond delocalizations generate a total interaction energy of 143.70 kcal/mol. In contrast, the C1–C2 σ bond, which lacks a π bond, only generates 34.98 kcal/mol from its 16 delocalizations. Compared to the typical delocalization in benzene (Table 3), the delocalization energies of 34.98 kcal/mol for the C1–C2 bond and 143.70 kcal/mol for the C1–C6 bond in fullerene C_60_ are approximately 38% higher than the corresponding 21.76 kcal/mol for the C1–C2 bond and 111.70 kcal/mol for the sum of the C1–C6 σ and π bonds in benzene. This increase is primarily due to the presence of six additional vicinal delocalizations for the C1–C2 bond, four additional geminal and six additional vicinal delocalizations for the C1–C6 bond in fullerene C_60_, as shown in Table 3. Notably, the C1–C6 π bond in fullerene C_60_ undergoes strong delocalization interactions with alternating antibonds, while in benzene, only two antibond acceptors interact with the C1–C6 Lewis-type donor bonds. Despite this, the delocalization energy of 25.66 kcal/mol for the C1–C6 bond in fullerene C_60_ is weaker than the 42.39 kcal/mol observed for the delocalization in benzene. For consistency, we have listed the results of the CCSD(T)/aug-cc-pVTZ calculations alongside those from MP2/6–31G* for benzene in Table 4 and Figure 5. The MP2/6–31G* results show a delocalization energy of 42.39 kcal/mol, closely matching the 41.94 kcal/mol obtained from CCSD(T)/aug-cc-pVTZ calculations for the strongest delocalization. Our calculations suggest that the C1–C2 and C1–C6 bonds in fullerene C_60_ contribute 38% and 22% more enthalpy, respectively, than those bonds in benzene, which may account for the thermochemical stability of fullerene C_60_ [14,49,50]. These findings imply that both non-planarity and formation entropy should be carefully considered when interpreting the thermostability of fullerene [52,79].

### 3.3. Correlation Energy and HOMO-LUMO Gap of α Electrons of Fullerene C_60_

Single-point MP2 and Hartree–Fock (HF) energies were calculated using various basis sets based on the optimized geometries. The energy discrepancies between MP2 and Recovering electron correlation energy is a fundamental and challenging task in quantum chemistry. Correlation energy typically refers to the difference between the true total energy and the Hartree–Fock limit. It consists of two main components: static and dynamic correlation [80]. Static correlation arises from the fact that HF treats the many-body wavefunction as a single Slater determinant, whereas the exact total energy requires a combination of many determinants in the non-relativistic Schrödinger equation [81]. Dynamic correlation, or Pauli correlation, results from the mean-field approximation in HF, where instantaneous Coulomb repulsion is approximated by treating other electrons as an average charge distribution. HF already accounts for static exchange correlation between parallel spin electrons based on the anti-symmetry principle. MP2, through its perturbative potential, recovers electron correlation by considering electron-electron interactions [82]. The MP2-HF energy difference has been widely applied in evaluating correlation energy and the electron density matrix [83]. The MP2-HF discrepancy provides a major component of the electron correlation energy in C_60_, which has a delocalized π bond system [84]. As shown in Table 5 and Figure 6, the MP2-HF energy differentials for fullerene C60 are −9.50, 13.65, and −13.49 kcal/mol per electron for calculations at MP2/3–21G*, MP2/6–31G, and MP2/Def2SVP, respectively. The MP2/Def2SVP basis set yields the same correlation energy per electron as MP2/6–31G*, while MP2/6–31G* shows a notable improvement of 4.15 kcal/mol per electron compared to MP2/3–21G*. A similar pattern is observed in the MP2-HF energy discrepancies for benzene across different basis sets. Notably, our calculations consistently show that fullerene C_60_ provides 2.30, 2.36, and 1.81 kcal/mol more correlation energy per electron than benzene at MP2/3–21G*, MP2/6–31G*, and MP2/Def2SVP, respectively. It is also worth mentioning that the CCSD(T)/aug-cc-pVTZ method produces significantly larger correlation energy (15.54 kcal/mol per electron) than all other results computed at the MP2 level, which motivates future CCSD(T) calculations on fullerene C_60_ derivatives. The importance of correlation energy is well-established in small delocalized π systems [85]. Aromaticity, a fundamental concept in organic chemistry [86], refers to the stabilization of conjugated cycloalkenes due to delocalized π-electrons following the 4n + 2 rule and exhibiting no diamagnetic properties in a magnetic field. The precise definition of aromaticity is still debated, but it has been evaluated from structural [87], energetic [88], reactivity [89], and magnetic perspective [90]. Fullerene C_60_ has been considered antiaromatic rather than aromatic due to its high positive heat of formation (610 kcal/mol) [12]. However, its aromaticity has also been evidenced both experimentally [18] and theoretically [16]. Magnetic shielding effects of frontier orbitals have been demonstrated in endohedral complexes of small molecules (e.g., helium or formaldehyde) encapsulated within fullerene C_60_ [19,60,61]. Therefore, other electronic factors influencing electron correlation in fullerene, such as strain destabilization energy, will be explored in future work [45].

The calculation of the LUMO energy for C_60_ has enabled the evaluation of designed fullerene derivatives for their potential use in organic photovoltaic applications [15,91] and reactivity studies [42]. According to quantum mechanics, the electron density of a perfect spherical molecule is confined to its surface. Spherical harmonic wavefunctions can be obtained by solving Schrödinger’s equation for such molecular systems. The energy levels corresponding to these harmonic states are determined based on the radius of the molecular system and angular momentum quantum numbers, following the Aufbau principle, Pauli exclusion principle, and Hund’s rule. Although fullerene C_60_ has an icosahedral symmetry rather than a perfect spherical geometry, its electronic energy scheme can be derived by expanding the energy level diagram for spherical symmetry harmonics with respect to angular momentum (l) [92]. The lower symmetry of the icosahedral group in C_60_ splits the degeneracies of the highest occupied molecular orbitals (HOMO) from spherical harmonic states [9,93], though a maximum molecular orbital degeneracy of five is still possible due to the icosahedral symmetry of fullerene C_60_. As shown in Figure 7, the HOMO of C_60_ has H_1u_ symmetry with a fivefold degeneracy, while the LUMO has a threefold degeneracy in T_1u_ symmetry. The LUMO+1 orbital, which is located just above the LUMO, also exhibits triply degenerate T_2u_ symmetry [94]. The transition from the HOMO to LUMO is optically forbidden due to the Ag symmetry of the ground state with icosahedral symmetry [13,44], so the HOMO-LUMO separation values in Table 6 are provided for reference. We focus on comparing the experimentally measured reduction potential with the HOMO to LUMO+1 transition. As shown in Table 6, the HOMO-LUMO+1 gap calculated using MP2/3–21G* is −8.94 eV, which is larger than the −8.66 eV gap obtained using MP2/6–31G*, suggesting that larger basis sets may yield more accurate HOMO-LUMO+1 gaps compared to the experimental value of 4.90 eV. The deviations from the experimental value are −4.04 eV and −3.76 eV for MP2/3–21G* and MP2/6–31G*, respectively [7]. In contrast, the HOMO-LUMO separations from the B3LYP/Def2SVP and BP86/Def2SVP calculations are 2.28 eV and 2.03 eV, respectively, which are lower than the experimental reduction potential. It is worth noting that MP2 calculations may yield more accurate results with larger basis sets. However, there remains significant room for improvement in the B3LYP/Def2SVP and BP86/Def2SVP calculations, as Def2SVP is already a basis set of reasonable size. Therefore, the results of the HOMO-LUMO+1 transition from B3LYP/Def2SVP and B3LYP/6-31G should be interpreted with caution in future work.

## 4. Discussion

Our calculations show relatively large deviations in the HOMO-LUMO gap compared to the available experimental value of −4.9 eV. Therefore, further studies on the HOMO-LUMO gap calculations for fullerene C60 will be pursued. Jaworski et al. reported that the state-of-the-art DLPNO-CCSD(T) method provides more accurate predictions for IR, Raman, UV-vis, ionization potential, NMR chemical shifts (^1^H and ^13^C), spin–spin coupling, and thermodynamic stability of fullerene, accounting for the effects of the structural Faraday cage [57]. To improve the accuracy of our results, we plan to apply higher-level methods, such as CCSD(T)/aug-cc-pVTZ, in future studies. These methods will be used to refine our HOMO-LUMO gap calculations and to explore delocalization, including bond–antibond interactions and Rydberg effects, in non-planar fullerene, in contrast to planar benzene, to better understand aromaticity.

## 5. Conclusions

The MP2 structural simulation of fullerene C_60_ confirms that its non-planarity is characterized by two types of dihedral angles—138° and 143°—in the optimized icosahedral structure. The 120 dihedral angles of 138° occur between two intersecting hexagon planes, while 60 dihedral angles of 143° are observed between a hexagon and a neighboring pentagon. The polyhedral structure has bond lengths of 1.42 Å for C–C bonds shared between two hexagons and 1.47 Å for C–C bonds connecting a hexagon and a pentagon. Our NBO analysis suggests that carbon atoms in the C–C bonds of 1.42 Å (shared by hexagons) exhibit sp^1.78^ hybridization, while those in the C–C bonds of 1.47 Å (connecting hexagons and pentagons) undergo sp^2.13^ hybridization. The ESP charge range for carbon atoms, from −0.04 a.u. to 0.04 a.u., represents a significantly larger magnitude than that of other charge types in fullerene C_60_. Second-order perturbation evaluations indicate that delocalization interaction energies involving C1–C6 and C1–C2 bonds in fullerene C_60_ are 22% and 38% higher, respectively, than the corresponding delocalization energies in planar benzene. These calculations highlight the significance of delocalization interactions in contributing to the formation of enthalpy and the thermostability of fullerene C_60_. Moreover, the major correlation energy per electron at MP2/Def2SVP for fullerene C_60_ yields a value of 13.49 kcal/mol, which is comparable to the 11.68 kcal/mol observed for benzene. This work establishes a solid benchmark foundation for future mechanistic studies of fullerene C_60_ derivatives for CO_2_ reduction systems.

## Figures and Tables

**Figure 1 nanomaterials-14-01576-f001:**
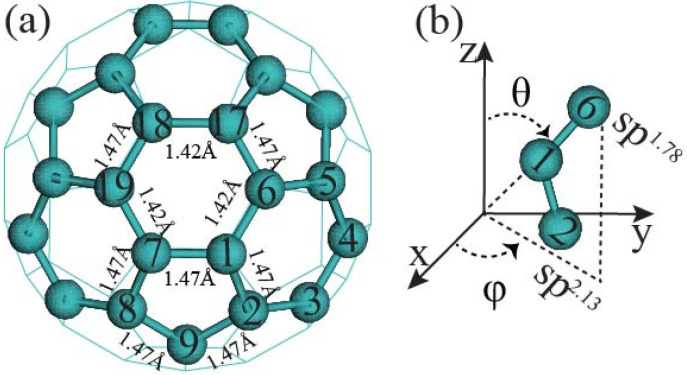
Structure of fullerene C_60_. (**a**) The hexagon and pentagon in CPK representation of fullerene C_60_; (**b**) The C1–C6 natural hybrid orbital (NHO) directionality described by a vector in a coordinating system. All figures are prepared using the visual molecular dynamics package [69].

**Figure 2 nanomaterials-14-01576-f002:**
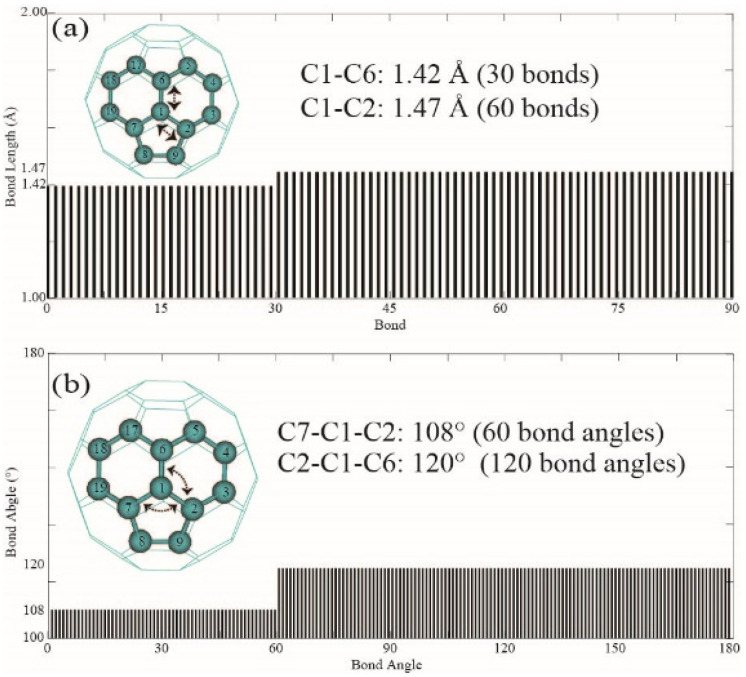
Profiles of bond lengths and bond angles of C_60_. (**a**) Bond lengths of C_60_; (**b**) Bond angles of C_60_.

**Figure 3 nanomaterials-14-01576-f003:**
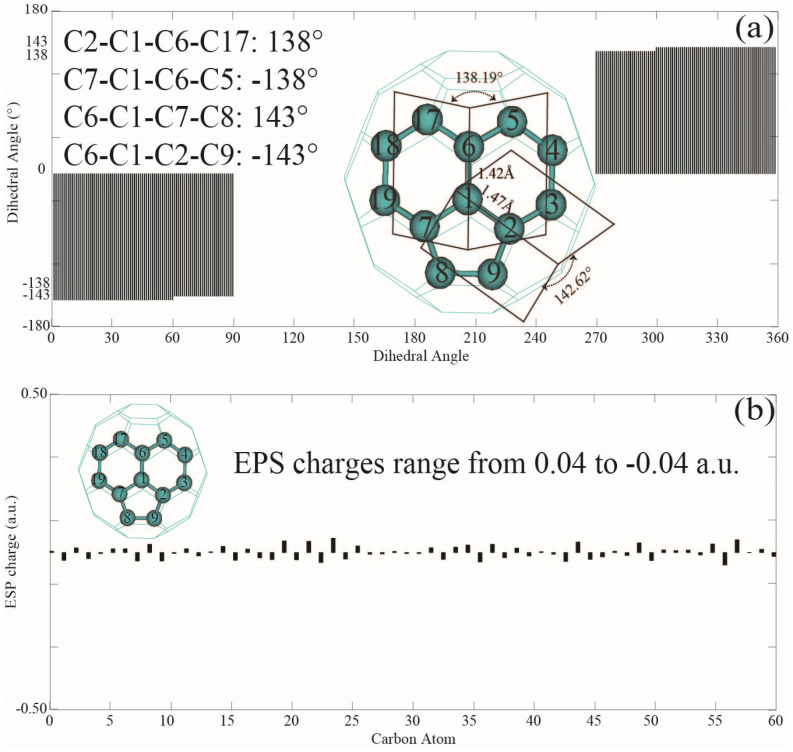
Dihedral angle and ESP partial charge of C_60_. (**a**) Dihedral angles of C_60_; (**b**) ESP charges of fullerene C_60_.

**Figure 4 nanomaterials-14-01576-f004:**
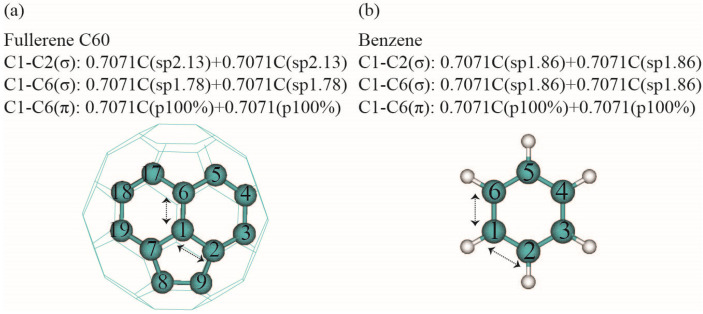
(**a**) Natural bond orbital coefficients and hybrids of fullerene C_60_; (**b**) natural bond orbital coefficients and hybrids of benzene.

**Figure 5 nanomaterials-14-01576-f005:**
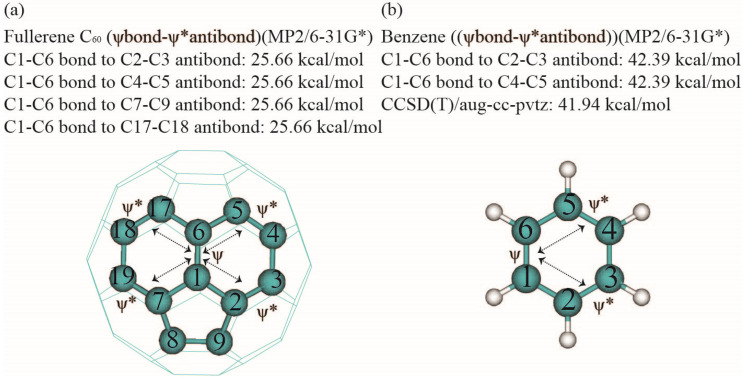
(**a**) Natural bond orbital delocalization of fullerene C_60_; (**b**) natural bond orbital delocalization of benzene. The star indicates antibond, the arrow indicates bond–antibond interaction.

**Figure 6 nanomaterials-14-01576-f006:**
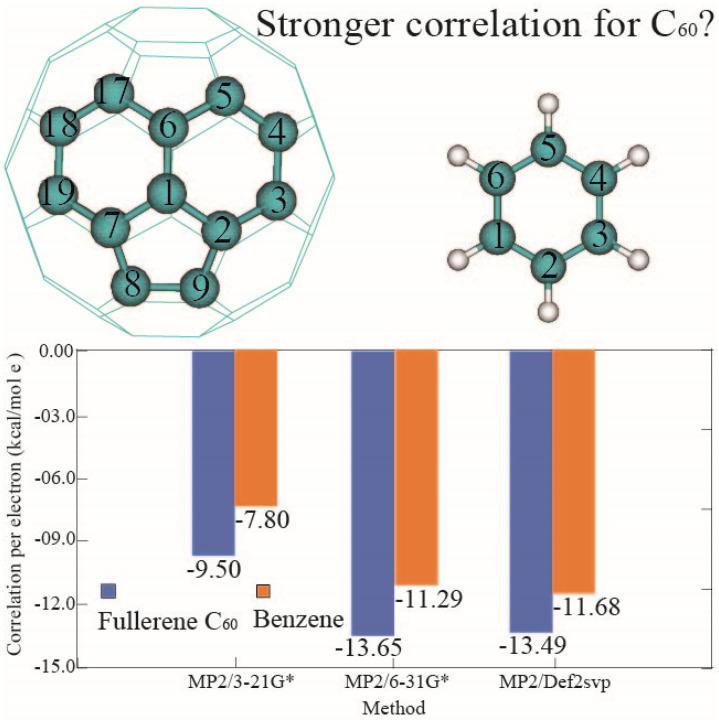
Comparison of correlation energy per electron between fullerene C_60_ and benzene.

**Figure 7 nanomaterials-14-01576-f007:**
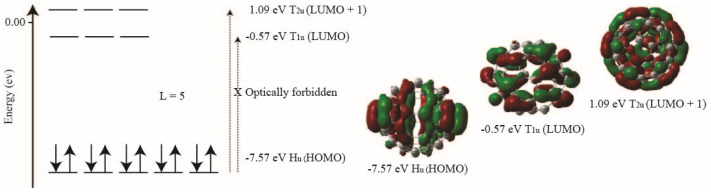
Energy level diagram of the L = 5 for fullerene C_60_ based on the results computed using MP2 method.

**Table 1 nanomaterials-14-01576-t001:** Ranges of partial charges of Fullerene C_60_.

Model	C1	C6	C2	Minimum	Maximum	Median
ESP	−0.031000	0.009000	−0.035000	−0.040000	0.040000	0.0
NBO	−0.000020	0.000010	0.000010	−0.000030	0.000030	0.0
Mulliken	0.000011	−0.000022	−0.000022	−0.000049	0.000046	0.0
Hirshfeld	−0.000020	0.000013	0.000010	−0.000022	0.000023	0.0
CM5	−0.000020	0.000013	0.000010	−0.000022	0.000023	0.0

**Table 2 nanomaterials-14-01576-t002:** Angular properties of natural hybrid orbitals.

Bond	Line of Center between Two Nuclei		Hybrid 1			Hybrid 2		
Hybrid 1–2	θ (Theta)	φ (Phi)	θ	φ	Dev	θ	φ	Dev
Fullerene C_60_								
C1–C2 (σ)	129.4	264.9	140.6	255.2	13.1	62.4	91.8	13.1
C1–C6 (σ)	47.9	167.4	58.7	154.1	15.1	140.8	5.5	15.1
C1–C6 (π)	47.9	167.4	50.3	276.1	75.8	71.9	297.4	75.8
Benzene								
C1–C2 (σ)	90.0	22.7	90.0	17.9	4.1	90.0	207.6	4.1
C1–C6 (σ)	90.0	142.7	90.0	147.6	4.1	90.0	317.9	4.1
C1–C6 (π)	90.0	142.7	0.0	0.0	90.0	0.0	0.0	90.0

**Table 3 nanomaterials-14-01576-t003:** Statistics of delocalization interactions larger than 0.5 kcal/mol.

Molecule	Bond	Energy (Hartree)	Geminal	Vicinal	Energy (kcal/mol)
Fullerene C_60_	C1–C2 (σ)	−0.88405	4	12	34.98
	Total	C1–C2	4	12	34.98 (100% + 38%)
	C1–C6 (σ)	−0.93273	4	8	26.33
	C1–C6 (π)	−0.34383	4	8	110.14
	Total	C1–C6	8	16	143.70 (100% + 22%)
Benzene	C1–C2 (σ)	−0.90988	4	6	21.74
	Total	C1–C2	4	6	21.76 (100%)
	C1–C6 (σ)	−0.91019	4	6	21.76
	C1–C6 (π)	−0.29735	0	4	87.66
	Total	C1–C6	4	10	111.70 (100%)

**Table 4 nanomaterials-14-01576-t004:** Delocalization energies for C1–C2 and C1–C6 bond of both fullerene and benzene using second order perturbation theory with MP2/631G* and/CCSD(T)/aug-cc-pvtz.

Fullerene C_60_				Benzene(MP2/6–31G*/CCSD(T)/aug-cc-pvtz)			
Donor	Type	Acceptor	E (kcal/mol)	Donor	Type	Acceptor	E (kcal/mol)
C1–C2 (σ)	Rydberg	C3	0.8	C1–C2 (σ)	Rydberg	C3	1.36/0.92
	Rydberg	C3	1.64		Rydberg	C3	2.02/1.93
	Rydberg	C6	0.8		Rydberg	C6	1.36/0.92
	Rydberg	C6	1.64		Rydberg	C6	2.02/1.93
	Rydberg	C7	1.31		Antibond	C1–C6	3.24/2.25
	Rydberg	C9	1.31		Antibond	C1–H7	1.49/0.91
	Antibond	C1–C6	3.91		Antibond	C2–C3	3.24/2.25
	Antibond	C1–C7	2.05		Antibond	C2–H8	1.49/0.91
	Antibond	C2–C3	3.91		Antibond	C3–H9	2.77/3.15
	Antibond	C2–C9	2.05		Antibond	C6-H12	2.77/3.15
	Antibond	C3–C11	3.48	Total	C1–C2	10	21.76/18.32
	Antibond	C6–C17	3.48	C1–C6 (σ)	Rydberg	C2	1.36/0.92
	Antibond	C7–C19	3.8		Rydberg	C2	2.02/1.93
	Antibond	C7–C19	0.5		Rydberg	C5	1.36/0.92
	Antibond	C9–C10	3.8		Rydberg	C5	2.02/1.93
	Antibond	C9–C10	0.5		Antibond	C1–C2	3.24/2.25
Total	C1–C2	16	34.98		Antibond	C1–H7	1.50/0.91
C1–C6 (σ)	Rydberg	C2	1.61		Antibond	C2–H8	2.77/3.15
	Rydberg	C5	1.61		Antibond	C5–C6	3.24/2.25
	Rydberg	C7	1.61		Antibond	C5–H11	2.77/3.15
	Rydberg	C7	1.61		Antibond	C6–H12	1.50/0.91
	Antibond	C1–C2	3.68		Total	10	21.78/18.32
	Antibond	C1–C7	3.68	C1–C6 (π)	Rydberg	C2	2.57/1.88
	Antibond	C2–C9	1.75		Rydberg	C5	2.57/1.88
	Antibond	C5–C6	3.68		Antibond	C2–C3	42.39/41.94
	Antibond	C5–C15	1.75		Antibond	C4–C5	42.39/41.94
	Antibond	C6–C17	3.68		Total	4	89.92/87.64
	Antibond	C7–C8	1.75				
	Antibond	C6–C17	1.75				
	Total	12	28.16				
C1–C6 (π)	Rydberg	C1	0.53				
	Rydberg	C2	1.75				
	Rydberg	C5	1.75				
	Rydberg	C6	0.53				
	Rydberg	C7	1.75				
	Rydberg	C17	1.75				
	Antibond	C1–C2	1.2				
	Antibond	C1–C7	1.2				
	Antibond	C2–C3	25.66				
	Antibond	C4–C5	25.66				
	Antibond	C5–C6	1.2				
	Antibond	C6–C17	1.2				
	Antibond	C7–C19	25.66				
	Antibond	C17–C18	25.66				
	Total	14	115.54				
Total	C1–C6	26	143.70 (122%)	Total	C1–C6	14	111.70 (100%)
Total	C1–C2	16	34.98 (138%)	Total	C1–C2	10	21.76 (100%)

**Table 5 nanomaterials-14-01576-t005:** The major correlation energy of fullerene C_60_ and benzene in addition to exchange component.

Method	MP2 (Hartree)	HF (Hartree)	MP2—HF (kcal/mol)	(MP2—HF)/ (kcal/mol e)
Fullerene C60				
MP2/3–21G*	−2264.4378913	−2258.9901941	−3418.48	−9.50
MP2/6–31G*	−2279.6317630	−2271.8025400	−4912.92	−13.65
MP2/def2svp	−2277.8316660	−2270.0928633	−4865.21	−13.49
Benzene (The CCSD(T)/aug-cc-pvtz recovered correlation is counted by CCSD(T)-HF)				
MP2/3–21G*	−229.9376703	−229.4155101	−327.657	−7.80
MP2/6–31G*	−231.4577321	−230.7018849	−474.302	−11.29
MP2/def2svp	−231.3162080	−230.5345726	−490.477	−11.68
CCSD(T)/aug-cc-pvtz	−231.8204089	−230.7804015	−652.615	−15.54

**Table 6 nanomaterials-14-01576-t006:** Comparison of the HOMO-LUMO gaps of α electrons for fullerene C_60_ using different methods (HOMO H_1u_, LUMO T_1u_, and LUMO + 1 T_2u_ α MOs).

Method (α Electron)	HOMO H_1u_ (eV)	LUMO T_1u_ (eV)	LUMO + 1 T_2u_ (eV)	H_1u_ to T_1u_ (eV)	H_1u_ to T_2u_ (eV)
MP2/3-21G*	−8.19	−0.92	0.75	−7.27	−8.94
MP2/6–31G*	−7.57	−0.57	1.09	−7.00	−8.66
Experimental [7]					−4.90
B3LYP/Def2svp	−7.63	−5.95	−5.01	−1.68	−2.62
BP86/Def2svp	−5.94	−4.13	−3.07	−1.81	−2.87

## Data Availability

The data are available from the corresponding author upon reasonable request.

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
