# Peer review of "A Quantum Mechanical MP2 Study of the Electronic Effect of Nonplanarity on the Carbon Pyramidalization of Fullerene C60"

_nanomaterials, 2024, doi:10.3390/nano14191576_

Round 1

Reviewer 1 Report

Comments and Suggestions for Authors

In the present manuscript, the authors report on the MP2/(dz basis set) geometry optimization of the C60 molecule. The authors also performed some NBO calculations and estimated the HOMO-LUMO gap and specific correlation energy. In my opinion, this contribution clearly lack novelty. The MP2/(TZ) geometry optimizations of C60 date back to 1991, surprisingly, not cited at all (10.1016/0009-2614(91)80301-D). There have been many other works on C60 employing similar techniques since that time, to name a few, 10.1016/S0009-2614(97)00733-1 10.1021/jp056027s

However, even more important point is that a plethora of modern reliable quantum chemical methods are applicable in the present case. The approach used by the authors does not meet the state-of-the-art quantitative computations (the correlated procedures with the basis sets smaller than TZ are seldom meaningful, a lot of DF, F12, local correlation approximations are available, etc.). See, for instance, 10.1039/D1CP02333K

Apart from this, the discussion section comprised of a single sentence looks weird.

Making the bottom line - I do not see in the manuscript any serious insight on the C60 geometrical and electronic structures worth publishing.

Author Response

In the present manuscript, the authors report on the MP2/(dz basis set) geometry optimization of the C60 molecule. The authors also performed some NBO calculations and estimated the HOMO-LUMO gap and specific correlation energy. In my opinion, this contribution clearly lack novelty. The MP2/(TZ) geometry optimizations of C60 date back to 1991, surprisingly, not cited at all (10.1016/0009-2614(91)80301-D). There have been many other works on C60 employing similar techniques since that time, to name a few, 10.1016/S0009-2614(97)00733-1 10.1021/jp056027s.

Responses: All the three works are cited in the introduction section. The reviewer is partially right about the novelty of this work. The work by Haser et al (https://doi.org/10.1016/0009-2614(91)80301-D) did the structural optimization up to TZP/MP2 level, but analysis was focused on the bond lengths and Raman frequencies (doi:10.1016/s0009-2614(97)00733-1). Our analysis makes comparisons of delocalization between nonpolar C60 and planar benzene, which provide more direction evidence of delocalization to aromatic dispersion. The interaction and NMR chemical shifts of noble gases in fullerene at MP2/tzp did not make the direct connections between nonplanar structural feature chemical shifts by Shameema et al(doi: 10.1016/S0009-2614(97)00733-1). The work (doi: 1 10.1021/jp056027s) used same MP2/6-31G* to study the blue shift of X-H stretching isoelectronic molecules within the fullerene cage not the any impact on the fullerene itself, but effect of delocalization on aromaticity was not discussed which is what we are focusing on.

However, even more important point is that a plethora of modern reliable quantum chemical methods are applicable in the present case. The approach used by the authors does not meet the state-of-the-art quantitative computations (the correlated procedures with the basis sets smaller than TZ are seldom meaningful, a lot of DF, F12, local correlation approximations are available, etc.). See, for instance, 10.1039/D1CP02333K.

Responses: The reviewer made the points about the availability of the plethora of reliable quantum mechanics methods but cost specifically the scratch space of almost over 1 TB for MP2/6-31G* could be a hurdle to many of the quantum community. The mentioned work by Jaworski et al (10.1039/D1CP02333K) presented the features of IR, Raman, UV-vis, ionization potential, NMR chemical shift of H1 and C13, spin-spin coupling as a result of effects of molecular Faraday cage and thermodynamical stability using the state-of-the-art DLPNO-CCST(T) other than MP2, SCS-MP2, and DPT-SAPT, whereas our MP2/6-31G* presents delocalization including bond-antibond interaction and Rydberg effect of nonplanar fullerene in contrast to those of planar benzene for the illustration of aromaticity. This clearly shows that two works are addressing interesting scientific topics from different perspectives, indicating the certain degree of novelty of our manuscript.

Apart from this, the discussion section comprised of a single sentence looks weird.

Responses: Application of higher-level methods such as DLPNO-CCST(T) are discussed in the section.

Making the bottom line - I do not see in the manuscript any serious insight on the C60 geometrical and electronic structures worth publishing.

Responses: I can hardly agree with this. This manuscript demonstrates unique and direct evidence of the effects of structural nonplanarity on the electronic aromaticity at lower computational cost which addresses an interesting concern to chemistry researchers. Since our results are never directly and clearly presented in the previous literature, including those mentioned by the reviewer, I would think that our manuscript deserves a more positive recommendation.

Reviewer 2 Report

Comments and Suggestions for Authors

Author present a work for studying the bond length, dihedral angles, and correlation energy of C60.  Some comments need to be addressed before publication.

1. Introduction: Author mentioned fullerene C60 derivative for CO2 reduction systems. I suggest more discussion or reference should be add in introduction to help readers to understand why the QM MP2 study is important.

2. C60 derivative as an electron acceptor is used for organic photovoltaics.  I suggest it should include "Materials Today 2012, 15 (12), 554-562" as reference in introduction.

3. For the DFT calculation, organic photovoltaics research also include the conjugated polymer system. I encourage to review "Chemical Physics Letters 2024, 850, 141447".

4. Check caption "Figure 6" is repeated twice. 

Comments on the Quality of English Language

No comment

Author Response

Comments and Suggestions for Authors

Author present a work for studying the bond length, dihedral angles, and correlation energy of C60.  Some comments need to be addressed before publication.

  1. Introduction: Author mentioned fullerene C60 derivative for CO2 reduction systems. I suggest more discussion or reference should be add in introduction to help readers to understand why the QM MP2 study is important.

Responses: More references are cited.

  1. C60 derivative as an electron acceptor is used for organic photovoltaics.  I suggest it should include "Materials Today 2012, 15 (12), 554-562" as reference in introduction.

The "Materials Today 2012, 15 (12), 554-562" is cited in the introduction.

  1. For the DFT calculation, organic photovoltaics research also include the conjugated polymer system. I encourage to review "Chemical Physics Letters 2024, 850, 141447".

Responses: The work "Chemical Physics Letters 2024, 850, 141447" is included in the references.

  1. Check caption "Figure 6" is repeated twice. 

Responses: The second Figure 6 is changed to Figure 7, and corresponding corrections are also made in the relevant section.

Reviewer 3 Report

Comments and Suggestions for Authors

The paper submitted by Yuemin Liu and Yunxiang Gao summarizes quantum calculations of C60 at MP2 level and discussed fundamental structural and electronic parameters such as bond lengths, angles, MO energy levels, NBO, and partial charges. All these information had been already well-examined in the past few decades. Therefore, I unfortunately could not find novel findings from the manuscript. However, I may believe that reconsideration of the known parameters at high calculations level that could not be achieved in the previous days might be meaningful in some area. Therefore, this paper could be accepted in Nanomaterials after addressing points raised below:

[General comment 1]

"C60" should be appropriately shown with subscripts in title and manuscript. 

[General comment 2] The sp characters should be appropriately shown with superscripts as seen in sp1.78 and sp2.13, for instance.

[General comment 3] Please use "eV" instead of "ev"

[Line 31-32] There are lack of recent experimental examples. Please include appropriate papers reported recently

[2. Materials and Methods] Please mention about symmetry restriction on optimization.

[Line 94, 106] Figure 1(d) is typo of Figure 1(b)?

[Line 122] change "indicatedin" to "indicated in"

[Line 124] The authors seem to talk that sp character is a pyramdization factor. However, this kind of discussion is not straightforward. Please discuss it more carefully. Pyramidization is a structural factor and it is unable to directly link to sp character. If the authors like to discuss pyramidization, the POAV angles must be discussed. The orbital directionality obtained from NBO analysis might helpful to show the relationship with pyramidization, though. 

[Line 197] As the authors already cited a paper reported by Hirsch, the aromaticity of fullerenes has been well-examined. In terms of experimental view, it has been well-understood experimentally by exploring shielding effect of endohedral species within C60. The authors should mention this point with appropriate references.

[Abstract and conclusion] I could not find any relationship between this work and CO2 reduction system though the authors mention "This work may aid in rational design of fullerene C60 derivative for CO2 reduction systems." Please add more careful and deeper discussion on how could the work be used to the rational materials design.

Author Response

Comments and Suggestions for Authors

The paper submitted by Yuemin Liu and Yunxiang Gao summarizes quantum calculations of C60 at MP2 level and discussed fundamental structural and electronic parameters such as bond lengths, angles, MO energy levels, NBO, and partial charges. All these information had been already well-examined in the past few decades. Therefore, I unfortunately could not find novel findings from the manuscript. However, I may believe that reconsideration of the known parameters at high calculations level that could not be achieved in the previous days might be meaningful in some area. Therefore, this paper could be accepted in Nanomaterials after addressing points raised below:

[General comment 1]

"C60" should be appropriately shown with subscripts in title and manuscript. 

Responses: The subscript format is applied to all terms in the title and manuscript.

[General comment 2] The sp characters should be appropriately shown with superscripts as seen in sp1.78 and sp2.13, for instance.

Responses: The superscript format is applied to all sp characters.

[General comment 3] Please use "eV" instead of "ev"

Responses: All the “ev”s were changed to “eV”s.

[Line 31-32] There are lack of recent experimental examples. Please include appropriate papers reported recently

Responses: The recent experimental works are discussed in line 31-32.

[2. Materials and Methods] Please mention about symmetry restriction on optimization.

 Responses: No symmetry restriction was applied.

[Line 94, 106] Figure 1(d) is typo of Figure 1(b)?

Responses: Correction of the error is made.

[Line 122] change "indicatedin" to "indicated in"

Responses: The mentioned typo is corrected.

[Line 124] The authors seem to talk that sp character is a pyramdization factor. However, this kind of discussion is not straightforward. Please discuss it more carefully. Pyramidization is a structural factor and it is unable to directly link to sp character. If the authors like to discuss pyramidization, the POAV angles must be discussed. The orbital directionality obtained from NBO analysis might helpful to show the relationship with pyramidization, though. 

Responses:  Normally, sp3 character indicates the pyramidal geometry and sp4 configuration correspondents to tetrahedral structure. Figure 1(b) described a vector for natural hybrid orbital in polar coordinating system, in which the θ and φ serve the same purpose as Haddon's pi-orbital axis vector (POAV) angles. The NHO angles are compared between nonplanar fullerene C60 and planar benzene.

[Line 197] As the authors already cited a paper reported by Hirsch, the aromaticity of fullerenes has been well-examined. In terms of experimental view, it has been well-understood experimentally by exploring shielding effect of endohedral species within C60. The authors should mention this point with appropriate references.

Responses: The work by Jarvis et al (https://doi.org/10.17635/lancaster/researchdata/483.) is cited shielding effect within fullerene.

[Abstract and conclusion] I could not find any relationship between this work and CO2 reduction system though the authors mention "This work may aid in rational design of fullerene C60 derivative for CO2 reduction systems." Please add more careful and deeper discussion on how could the work be used to the rational materials design.

Responses: The discussion of CO2 reduction is made based on the original research by Meloni et al (https://doi.org/10.3389/fchem.2021.712960), Andreoli et al ( 

https://doi.org/10.1002/cssc.201500605), and Ridassepi et al (https://doi.org/10.1021/acs.jpcc.4c01468) in the introduction.

Round 2

Reviewer 1 Report

Comments and Suggestions for Authors

I am not satisfied with authors' reply. The NBO calculations are very trivial, the discussion on the Table 1 (with six decimal points!!) is meaningless, all these charges are actually zero (yes, 0.03 too). The HOMO-LUMO gap calculations attempted by the authors are actually at the HF level, the MP2 does not affect orbitals and only generates the energy correction to the ground state. Other correlated methods also employ the HF orbitals (this statement is entirely meaningless: "Therefore, higher level methods than MP2/6-31G* will be applied in future studies of HOMO-LUMO gap"). The difference between their values are entirely due to different basis sets employed in the HF calculations. Yes, HF is well-known to overestimate the bandgap, due to mostly "anionic" nature of the virtuals. Please, consult any good QC textbook (e.g., Jensen). Finally, it is not entirely clear, what "the experimental HOMO-LUMO gap" is, most like, the electrochemical reduction potentials. They do not correspond to the orbital gap quantitatively, as the electron-electron interaction in the HF theory is self-consistent. Next, the absolute specific correlation energy does not have any important meaning, this values will be entirely different at another correlated level (e.g., CCSD). If the authors really wanted anything reasonable, this could be a contribution of different levels of correlation to total atomization energies.

The literature on the the C60 calculations is super vast. Please, get acquainted with it properly before attempting to reinvent the bicycle.

Author Response

Comments and Suggestions for Authors

I am not satisfied with authors' reply. The NBO calculations are very trivial, the discussion on the Table 1 (with six decimal points!!) is meaningless, all these charges are actually zero (yes, 0.03 too). The HOMO-LUMO gap calculations attempted by the authors are actually at the HF level, the MP2 does not affect orbitals and only generates the energy correction to the ground state. Other correlated methods also employ the HF orbitals (this statement is entirely meaningless: "Therefore, higher level methods than MP2/6-31G* will be applied in future studies of HOMO-LUMO gap"). The difference between their values are entirely due to different basis sets employed in the HF calculations. Yes, HF is well-known to overestimate the bandgap, due to mostly "anionic" nature of the virtuals. Please, consult any good QC textbook (e.g., Jensen). Finally, it is not entirely clear, what "the experimental HOMO-LUMO gap" is, most like, the electrochemical reduction potentials. They do not correspond to the orbital gap quantitatively, as the electron-electron interaction in the HF theory is self-consistent. Next, the absolute specific correlation energy does not have any important meaning, this values will be entirely different at another correlated level (e.g., CCSD). If the authors really wanted anything reasonable, this could be a contribution of different levels of correlation to total atomization energies.

Responses: We are intent to understand the aromaticity of C60 from delocalization bond-antibond in the NBO analysis. The NBO analysis might be trivial according to reviewer probably because of the limitations of NBO theory, but the topic of aromaticity is not trivial. Our NBO analysis of aromaticity is therefore not trivial even if it is not perfect or very accurate. Partial charge is one of the common electronic parameters, we are showing that the larger magnitude of ESP charge could serve as an indicator for structural effect for future studies of doped fullerene. The HOMO-LUMO gaps are entirely due to different basis sets employed in the HF calculations probably because of MP2 density was not used, and this is illustrated now in the paragraph. Additionally, the correlation forces is likely very close between HOMO and LUMO, so MP2 HOMO-LUMO gap does not make much difference to HF HOMO-LUMO discrepancy. The CCSD(T)/aug-cc-pvtz calculations are done for benzene, and discussions for fullerene are included in the manuscript now. We are working on the CCSD(T)/aug-cc-pvtz computation of fullerene.

The literature on the the C60 calculations is super vast. Please, get acquainted with it properly before attempting to reinvent the bicycle.

Responses: We tried to cite the relevant works as completely as possible from the huge number of literatures depending on our simulations.

Reviewer 3 Report

Comments and Suggestions for Authors

The revised manuscript includes some changes while the responses toward one question are not fully convincing. Although the authors cited some references, these are not related to those I suggested.

[Line 197] As the authors already cited a paper reported by Hirsch, the aromaticity of fullerenes has been well-examined. In terms of experimental view, it has been well-understood experimentally by exploring shielding effect of endohedral species within C60. The authors should mention this point with appropriate references.

Responses: The work by Jarvis et al (https://doi.org/10.17635/lancaster/researchdata/483.) is cited shielding effect within fullerene.

Comments: Because of the revision, it might be anymore Line 197. But the authors mentioned fullerene aromaticity with important review paper by Hirsch as ref. 58. This reference describes concepts of fullerene aromaticity. As ref. 16, the authors cited an experimental work. However, this paper discussess "chemical shielding" but not "magnetic shielding" that the authors focus on. Please cite appropriate references. There are many papers about magnetic shielding as found for He@C60, H2O@C60, H2@C60, H2O@C59N, ...

Author Response

Comments and Suggestions for Authors

The revised manuscript includes some changes while the responses toward one question are not fully convincing. Although the authors cited some references, these are not related to those I suggested.

[Line 197] As the authors already cited a paper reported by Hirsch, the aromaticity of fullerenes has been well-examined. In terms of experimental view, it has been well-understood experimentally by exploring shielding effect of endohedral species within C60. The authors should mention this point with appropriate references.

Responses: The work on experimental evidence for aromaticity of fullerene C60 by Stemfeld et al (https://doi.org/10.1021/ja012649p) is cited for the experimental evidence. 

Comments: Because of the revision, it might be anymore Line 197. But the authors mentioned fullerene aromaticity with important review paper by Hirsch as ref. 58. This reference describes concepts of fullerene aromaticity. As ref. 16, the authors cited an experimental work. However, this paper discussess "chemical shielding" but not "magnetic shielding" that the authors focus on. Please cite appropriate references. There are many papers about magnetic shielding as found for He@C60, H2O@C60, H2@C60, H2O@C59N, ...

Responses: The references (https://doi.org/10.1016/0009-2614(94)00844-2, DOI: 10.1039/D3CP00256J, and https://doi.org/10.1038/s41467-024-46886-5 ) regarding shielding effect of frontier orbitals of fullerene on endohedral small molecules is listed in the paragraph for comparison.

Round 3

Reviewer 3 Report

Comments and Suggestions for Authors

Although some important references on pioneering works (based on experiments) related to this manuscript were suggested twice to be included, the authors seem to avoid including them. The authors added four more references in the last revision. However, half of them are still computational works. Anyway, I believe inclusion of the two experimental works in the last revision makes readers wider in field. Therefore, I would like to suggest this manuscript to be accepted in this journal.